# Benefits of Physical Activity in Children with Cardiac Diseases—A Concise Summary for Pediatricians

**DOI:** 10.3390/children11121432

**Published:** 2024-11-26

**Authors:** Alina Costina Luca, Elena Țarcă, Valentina-Georgiana Tănase, Ioana-Alexandra Pădureț, Teodora-Simina Dragoiu, Lăcrămioara Ionela Butnariu, Solange Tamara Roșu, Iulia Cristina Roca, Dana-Elena Mîndru

**Affiliations:** 1Department of Pediatrics, Faculty of Medicine, “Grigore T. Popa” University of Medicine and Pharmacy, 700115 Iasi, Romania; alina.luca@umfiasi.ro (A.C.L.); mindru.dana@umfiasi.ro (D.-E.M.); 2Department of Surgery II, Discipline of Pediatric Surgery, “Grigore T. Popa” University of Medicine and Pharmacy, 700115 Iasi, Romania; 3Sfânta Maria’ Emergency Children’s Hospital, 700309 Iasi, Romania; georgiana_valentina1@yahoo.com (V.-G.T.); paduret.alexandra@gmail.com (I.-A.P.); 4Department of Sports Medicine, “Carol Davila” University of Medicine and Pharmacy, 020021 Bucharest, Romania; teodora-simina.ionescu@rez.umfcd.ro; 5Department of Medical Genetics, Faculty of Medicine, “Grigore T. Popa” University of Medicine and Pharmacy, 700115 Iasi, Romania; ionela.butnariu@umfiasi.ro; 6Department of Nursing, “Grigore T. Popa” University of Medicine and Pharmacy, 700115 Iasi, Romania; rosusolange@yahoo.com; 7Department of Surgery II, Discipline of Emergency Medicine, “Grigore T. Popa” University of Medicine and Pharmacy, 700115 Iasi, Romania; iulia.roca@umfiasi.ro

**Keywords:** congenital cardiac malformations, physical activity, cardiovascular system, children

## Abstract

A physically active lifestyle offers multiple benefits, including lowering the risk of cardiovascular disease, lowering body-mass index (BMI), and, last but not least, improving the quality of life. However, there are still disincentives to physical activity in children with heart diseases due to the high protection of parents and the scarcity of data in the literature. The purpose of this paper is to help pediatricians and pediatric cardiologists identify the type of physical activity allowed in children with congenital cardiac malformations, thus minimizing the risk of major adverse effects, such as acute coronary syndrome and sudden cardiac death. Therefore, we searched various electronic databases, such as PubMed, ScienceDirect, and Embase. We selected 61 articles published between 2008–2024. These articles included data on pediatric patients, from newborn to adolescent age. We decided to choose the 2008 study because of its focus on the essential role of education in schools regarding physical activity and the prevention of complications from sedentary lifestyles. Subsequently, we analyzed the data available in the literature up to 2024 regarding the type, intensity, and duration of exercise for patients with various congenital heart malformations. The conclusions of this review are presented based on the category of heart disease. There are differences in the free practice of sports in children with cardiomyopathies, atrioventricular block, arrhythmias with a genetic substrate, valvulopathies, and cyanogenic and non-cyanogenic congenital malformations. For example, children with arrhythmogenic right ventricle cardiomyopathy are not allowed to participate in sports competitions, but they can perform low physical activity for 150 min/week—golf, table tennis, and photography. However, it is recognized that a physically active lifestyle correlates with a decreased risk of cardiovascular diseases, body mass index, and an improvement in the quality of life. Children with congenital heart disease who are active have improved their cardiovascular systems.

## 1. Introduction

Congenital heart disease is a term used to describe developmental abnormalities of the heart, vessels, or both. There are many types of congenital heart disease, which vary in complexity, so the classification mild/medium/severe has been proposed [1]. It is estimated that approximately 9 out of 1000 newborns are affected, but thanks to advances in medical technology, more than 97% of them reach adulthood [2]. However, a percentage of patients show reduced physical and mental functioning and a reduced quality of life. In order to improve them, surgeries have been performed over the last two decades to treat and increase survival rates. Cardiac surgery in patients with congenital malformations has been linked to impaired hemodynamics and cognitive function. Other complications include renal dysfunction, restrictive lung disease, anemia, and a reduced social life. In this regard, multi-management is becoming increasingly more important for these patients today [3,4,5]. Thus, the American Heart Association and the European Society of Cardiology have proposed a training program with physical exercises postoperatively. It is considered beneficial due to the following: it improves cardiac function, regulates regenerative capacity, reduces the inflammatory response, and increases the quality of life [6].

An important point raised in recent studies is the tendency for children with congenital heart conditions to become less active as they age. This decline may result from overprotective parenting or cultural and societal expectations that limit the physical capabilities of these children. While parents’ concerns are understandable, they often underestimate their children’s abilities and, in doing so, inadvertently contribute to decreased activity levels and a higher risk of cardiovascular disease in the future. It is essential for healthcare providers to communicate effectively with families, emphasizing the potential benefits of physical activity and providing specific, tailored advice.

Some studies show significant results for improved mental state and exercise capacity. To obtain these results, a program of regular physical activities is necessary to avoid the harmful effects of a sedentary lifestyle. In addition, it is also associated with a reduced risk of obesity and ischemic heart disease. However, only a percentage of patients are encouraged to exercise, probably due to uncertainty about the effects, and so these children are more likely to be overweight than healthy ones [7,8]. For example, Swan and Hillis conducted a survey involving 99 young adults attending a pediatric and adolescent congenital outpatient clinic, revealing that relatively few patients (19%) were encouraged to exercise by their physician. Additionally, advice from physician assistants was often prohibitive (30%) despite the fact that most patients had simple and hemodynamically non-significant lesions. Similarly, Dean et al. reported that 40 out of 177 adolescents and young adults with congenital heart disease (22.5%) faced restrictions from participating in at least one sport due to their doctor’s recommendations [9].

To this date, there have been few studies focusing on the effects of exercise programs, so this article has a particularly important purpose, namely, to synthesize an update of information regarding physical activity recommendations for children with congenital heart defects. The 2020 Guidelines on Sports Cardiology and Exercise in Patients with Cardiovascular Disease are the cornerstone in this area. This edition contains updated information and recommendations on physical activity in both healthy and affected populations. The two major adverse events are acute coronary syndrome and sudden cardiac death [10].

We all know the benefits of a physically active lifestyle, both for the health and the quality of life of patients with these pathologies. The European Society of Cardiology considers that 60 min of moderate physical activity per day in children with cardiac malformations is needed. The American Association also does not have evidence to support restricting physical activity. However, it is crucial to understand that significant restrictions exist depending on the underlying pathology [11].

The purpose of this paper is to identify the type of physical activity allowed in children with congenital cardiac malformations, thus minimizing the risk of major adverse effects, such as acute coronary syndrome and sudden cardiac death.

## 2. Materials and Methods

For this narrative review, we searched for different studies in electronic databases, such as PubMed, ScienceDirect, and Embase. We selected 61 narrative reviews, guidelines, and original articles focusing on studies published between 2008–2024, which included data on pediatric patients ranging from newborns to adolescents. Following this, we analyzed the literature available up to 2024 concerning the type, intensity, and duration of exercise for patients with various congenital heart malformations.

Search terms about congenital cardiac malformations (e.g., “heart defect”, “cardiomyopathies”, “Fallot Tetralogy”, “valvulopathies”, “cardiomyopathies”, atrial septal defect, ventricular septal defect) were used independently and also in combination with terms about physical activity (e.g., “exercise”, “activity”, “physically active lifestyle”, etc.) and children (e.g., “youth”, “child”).

The articles were thoughtfully sorted, and duplicates were removed. Afterward, the references were searched for additional literature.

## 3. Results

### 3.1. Current Guidelines

The electrophysiological and hemodynamic parameters were the basis of physical activity recommendations in adolescents with cardiac malformations, according to the European Cardiology Society and the Sport Cardiology Guide. The evaluation of these children was carried out in stages and was based on five parameters: the pressure of the pulmonary artery, the existence of arrhythmias, the ventricular function, the size of the aorta, and the partial oxygen arterial saturation both at rest and during effort. Each patient will receive an individual prescription of exercises, adapted to the level of intensity allowed by their pathology [12].

The classification of Mitchell has been updated so that sports are now classified according to the short- and long-term impact of sport on cardiac morphology and according to the existence of hemodynamic changes, associated or not with training. In this sense, sports are now divided into four main categories: skill, power, mixed, and endurance. It is worth mentioning that some sports can cross different categories; for example, table tennis has a significant dynamic component but is a skill sport. Also, other factors must be taken into account, such as the emotional, psychological profile, and environmental conditions [12,13].

The heart rate, blood pressure values, and the long-term impact on cardiac remodeling and cardiac flow are the most important parameters underlying the division of sports disciplines. In skill exercises, there is no cardiac remodeling. This category includes the following: archery, golf, motor racing, horse riding, sailing, diving, photography, ski jumping, and table tennis. Strengthening sports are performed with low static effort, and performance depends on muscle strength. During repeated exercises, the values of blood pressure and heart rate will increase considerably. This will lead to cardiac remodeling and, as a result, the increase in the thickness of the left ventricular wall and, at the same time, the growth of the left ventricular cavity. Examples: shot-putting, snowboarding, sprinting, water skiing, wrestling, alpine skiing, and bobsleigh. Mixed sports refer to the alternation of dynamic work phases and static and recovery work. Examples in this category are ball sports and team sports. There are differences in the intensity of the exercise and the duration of the exercise, which vary depending on the individual’s role in the respective sport and depending on the type of sport. As was mentioned above, cardiac remodeling occurs with the increase in the size of the left ventricular cavity or with the modification of the thickness of the left ventricular wall. Examples: basketball, football, handball, fencing, rugby, tennis, and volleyball. The last category describes endurance sports. These are characterized by prolonged and intensive dynamics, often associated with static exercises and increased cardiac flow, increasing cardiac frequency and blood pressure values over several hours. In this category, cardiac remodeling is present, along with a significant increase in the size of the left ventricular cavity and the thickness of the wall. This category includes boating, cycling, medium-long swimming, running, and skating [12,13].

Sports can also be classified according to the characteristics of the exercise: frequency, intensity, duration, type (static—Rugby scrum, arm-wrestling, rock-climbing/dynamic—jogging, running, hockey, football, etc.), and mode of exercise (metabolic/muscular). The classification according to intensity is low-intensity, moderate-intensity, and high-intensity sports. According to the ESC Guidelines, each one has certain characteristics—low intensity: VO_2_ max (%) < 40, HRmax (%) < 55, HRR (%) < 40; moderate intensity: VO_2_ max (%) = 40–69, HRmax (%) = 55–74, HRR (%) = 40–69; high intensity: VO_2_ max (%) = 70–85, HRmax (%) = 75–90, HRR (%) = 70–85; very high intensity: VO_2_ max (%) > 85, HRmax (%) > 90, HRR (%) > 85. VO_2_ max = maximum oxygen consumption; HRmax = maximum heart rate; HRR = heart rate reserve. Examples: low-intensity exercises—golf, table tennis, shooting, curling, bowling, alpine skiing (recreational), basketball (adapted), handball (adapted); moderate-intensity exercises—sailing, judo, karate, alpine skiing, volleyball, mid/long distance running, dancing; high-intensity exercises—wrestling, boxing, rugby, hockey, fencing, basketball (competitive), handball (competitive), cycling (road), rowing, canoeing [12,14].

It is also important to remember the distinction between recreational and competitive sports: ESC defines a competitive athlete as a person who engages in regular physical exercises and takes part in official competitions.

The American Heart Association defines a person who participates in competitive sports as someone engaged in daily, periodic, harmonious training with high intensity. As is commonly known, sports for pleasure and leisure activities are part of recreational sports [15].

### 3.2. General Nutritional Aspects in Supporting Physical Effort

Numerous studies have highlighted the impact of lifestyle and nutrition on heart diseases in pediatric populations. The World Health Organization (WHO) employs various Z scores to assess children’s growth and nutritional status, categorizing them into underweight, normal weight, overweight, and obesity based on BMI. Obesity and malnutrition significantly affect health and quality of life, increasing mortality risk and contributing to cardiovascular, endocrine, and metabolic diseases, as well as posing challenges in school performance [16].

Children with congenital heart disease commonly experience malnutrition, which can severely affect bodily functions, including myocardial and pulmonary performance, and lead to recurrent infections due to weakened immune responses. Feeding protocols are crucial for these children, especially those on diuretics, who may suffer from electrolyte imbalances [16,17].

Evidence-based guidelines are designed to improve the nutritional status and health outcomes of these children. A balanced diet, alongside regular physical activity, is essential for optimal growth, cognitive development, and overall well-being while helping to mitigate complications associated with congenital heart malformations. Sports nutrition plays a vital role in promoting children’s health [18].

### 3.3. Features in the Indications of Physical Effort in Patients with Cardiovascular Disease

The main characteristics, the usual treatment recommended for each individual heart disease, as well as the type of suitable physical activity, are systematized in Table 1.

#### Adapted/Adaption of Specific Sports-Graded Exercise Program

Criteria for Defining Low- to Moderate-Intensity Physical Activity and Long-term Exercise [15]

Low-Intensity Physical Activity:○Heart Rate: Generally below 50–55% of maximum heart rate (calculated as 220 minus age).○Perceived Exertion: A Rate of Perceived Exertion (RPE) of 6–10 on a scale from 6 (no exertion) to 20 (maximal exertion).○Examples: Walking, light gardening, casual cycling, light household chores.○Duration: This can vary, but typically, longer sessions (30–60 min) are common in this category.Moderate-Intensity Physical Activity:○Heart Rate: Between 50–70% of maximum heart rate.○Perceived Exertion: An RPE of 11–13.○Examples: Brisk walking, dancing, recreational swimming, or light jogging.○Duration: 20–60 min per session, usually more than three to five days per week.Long-Term Exercise:○Definition: Exercise that is performed consistently over months to years and is adapted for cardiovascular, muscular, and metabolic improvements.○Intensity: This can range from moderate to vigorous intensity, depending on the goal (e.g., maintaining cardiovascular health or building endurance).○Recommended Guidelines (e.g., American College of Sports Medicine, 2023):□At least 150 min per week of moderate-intensity exercise or 75 min of vigorous-intensity exercise.□This can be broken into 30-min sessions on most days of the week.□For children and adolescents, 60 min of physical activity daily is recommended, with a mix of aerobic, muscle-strengthening, and bone-strengthening activities.

Low-intensity activities like walking, light swimming, and non-competitive sports (e.g., casual cycling, recreational dance) are usually encouraged for children with mild or repaired CHD, provided they are symptom-free. Moderate-intensity activities may be allowed in children post-repair (e.g., after surgery for ASD/VSD) as long as they are monitored and do not present significant risks. High-intensity activities like competitive sports (e.g., basketball, football) are generally not recommended for children with unrepaired or severe CHD unless specifically cleared by their cardiologist. After successful surgical repair, many children can gradually return to competitive sports once they demonstrate an appropriate level of fitness and are symptom-free.

### 3.4. Cardiomyopathies

#### 3.4.1. Hypertrophic Cardiomyopathy

Hypertrophic cardiomyopathy is a heart muscle disease characterized by abnormal thickening of the muscle tissue of the left ventricle of the heart. This occurs in the absence of an alternative cause, such as high blood pressure or aortic stenosis. Most cases are due to mutations in the sarcomeric protein. However, the specific mutation often remains indefinite. There are patients who remain asymptomatic, but there can also be cases of sudden cardiac death. It is well known that it is the most frequent cause of sudden death in young people [19].

In 25% of patients, the dynamic obstruction of the ejection tract of the left ventricle is detected due to left ventricular hypertrophy (LVH). In individuals with left ventricular hypertrophy (LVH), especially those diagnosed with hypertrophic cardiomyopathy, the left ventricular outflow tract may narrow during systole due to increased muscle mass and hyperdynamic contraction. This dynamic obstruction leads to heightened pressure gradients within the outflow tract, resulting in significant physiological changes. A critical risk associated with this obstruction is the development of ventricular arrhythmias, which can result in syncope or sudden cardiac arrest. The combination of impaired filling of the ventricles, rapid heart rates, and increased wall stress can trigger lethal arrhythmias such as ventricular tachycardia or ventricular fibrillation [51].

The treatment includes the prevention of sudden death, with treatment modalities focused on lifestyle changes pharmacotherapies, and in some patients with significant obstruction of the ejection tract, it may be necessary to implant a bicameral cardiac stimulator. The purpose is to reduce dynamic obstruction [52].

In previous studies, experts have considered that the restriction from sports competitions of athletes with hypertrophic cardiomyopathy is beneficial. However, several clinical studies have recently shown that the risk of sudden death in these patients is lower than previously thought. Several cohort studies have concluded that there is little evidence that all patients with hypertrophic cardiomyopathy suffer from activity-induced arrhythmias [52].

Evaluation of these patients is essential and should include both the personal history of exercise prior to diagnosis and the presence of other risk factors for sudden cardiac death. Functional capacity is assessed using ergometry in patients with hypertrophic cardiomyopathy who wish to perform physical movement [52].

After a complete assessment of the patient, the physician must take into account several factors in recommending physical activity, including symptoms, ESC score, which measures the risk of sudden death, and hemodynamic response to physical exertion, as well as the presence of arrhythmias both at rest and after exertion [20].

In conclusion, patients with hypertrophic cardiomyopathy, according to the SEC Guide for Sports Cardiology, have the following recommendations: participation in sports competitions or increased intensity physical activity (shot-putting, snowboarding, sprinting, water skiing, wrestling, alpine skiing), except syncope which can cause death, can be performed by patients who have no increased risk marker; participation in low and moderate intensity recreational physical activity (golf, motor racing, horse riding, sailing, diving, photography, ski jumping, table tennis) can be performed by patients with increased risk markers; participation in all competitions can be performed by individuals with positive gene mutations but who have a negative phenotype [20].

It is important to consider the annual follow-up of those practicing physical activities on a regular basis and a follow-up every six months for adolescents and young people who are more prone to sudden physical exercise-induced death [20].

#### 3.4.2. Arrhythmogenic Right Ventricular Cardiomyopathy

Arrhythmogenic cardiomyopathy of the right ventricle is a rare hereditary disease. It is characterized by the progressive replacement of fatty or fibrofatty myocytes, and this can lead to arrhythmias, such as ventricular ones, but also sudden cardiac death. They occur in an early stage. In advanced forms, it leads to right or biventricular insufficiency. The disease is most often determined by genetic factors and is caused by mutations in genes involved in the coding of structural cardiac proteins. Symptoms include ventricular arrhythmia, syncope, or even sudden death, but most patients are asymptomatic. Treatment is based on beta-blockers, but life-threatening arrhythmias may require the implantation of a cardiac defibrillator [21].

Patients with damage to the left ventricle represent only a minority. Thus, the guidelines in the current literature may not accurately reflect the impact of sports on these patients, as in the case of those with arrhythmogenic right ventricular cardiomyopathy [21].

Study results show that in patients participating in sports competitions, symptoms appeared earlier, once the physical activity was initiated at a younger age, and the reduction in exercise intensity is associated with a reduced risk of ventricular tachyarrhythmias or even death [21].

As with hypertrophic cardiomyopathy, initial assessment is essential, and exercise capacity is assessed by the exercise test. Of great importance is the syncope produced by arrhythmias, which is a major risk factor for sudden cardiac death and requires cardiac defibrillator therapy. Therefore, those with a history of syncope and cardiac arrest, as well as those with exertional symptoms, are advised to perform only recreational and low-intensity physical activity—VO_2_ max (%) < 40, HRmax (%) < 55, HRR(%) < 40—golf, table tennis, shooting, curling, bowling, alpine skiing (recreational), basketball (adapted), handball (adapted) [11].

In conclusion, patients with arrhythmogenic cardiomyopathy are advised to perform physical activity as follows: all could perform low-intensity exercise for 150 min per week—golf, table tennis, and photography; participation in competitive physical activity is not allowed [20].

As in the case of hypertrophic cardiomyopathy, annual follow-up is recommended for all who perform regular physical exercise. In adolescents and young people who are more prone to exercise-induced sudden cardiac death, 6-month follow-up is recommended. Six-month follow-up is also recommended for patients with a high arrhythmogenic risk genotype [20].

#### 3.4.3. Dilated Cardiomyopathy

Dilated cardiomyopathy is characterized by left or biventricular dilation and systolic dysfunction. Pathogenesis involves multiple mutations of several genes, but there are also non-genetic forms caused by predominantly viral infections that can lead to inflammation of the heart muscle, certain drugs or allergens, autoimmune diseases, or pathologies of the endocrine system. As with other cardiomyopathies, the dilated one may manifest in the form of sudden death but also heart failure, arrhythmias, embolism, or cardiac conduction. Echocardiography is essential and necessary because it evaluates the degree of ventricular dilation and myocardial remodeling. When the doctor identifies infection as a cause, histopathological analysis of an endomyocardial biopsy sample is indicated [22].

Standard management is based on the treatment of heart failure, but first of all, prevention is essential because dilated cardiomyopathy will eventually lead to impaired contractility. In some cases, certain specific therapies may be required, such as implantable defibrillators, namely in the case of life-threatening arrhythmias [23].

Evaluation of patients with dilated cardiomyopathy is essential and should include identification of etiology, history of prior physical activity, and functional capacity using an exercise test (ergometry); evaluation of the degree of left ventricular dilation, left ventricular dysfunction, and hemodynamic response to exercise; clinical presentation, and evaluation of exercise-induced arrhythmias [11,23].

It is not ethical to encourage symptomatic people to participate in sports competitions or in recreational sports with moderate intensity. Asymptomatic patients with mild left ventricular dysfunction or no arrhythmias during exercise could participate in competitive sports, e.g., basketball, football, handball, fencing, rugby, soccer, tennis, and volleyball [11,23].

Physical activity improves ventricular function, functional capacity, and, last but not least, the psychosocial quality of life in these patients. Therefore, exercise with variations in intensity should be part of the therapeutic management. However, it is extremely important to mention that vigorous exercise could be a cause of sudden cardiac death in dilated cardiomyopathy [20].

Annual evaluation of patients with this pathology performing regular exercise is recommended by current guidelines. A six-month evaluation should be performed in patients with high-risk mutations and in adolescents predisposed to sudden death associated with exercise [20].

### 3.5. Atrioventricular Block

Congenital complete atrioventricular block is a rare heart disease with an estimated incidence of 1 in 20,000 live births, and in most cases, it occurs after damage to the fetal heart by maternal autoantibodies against ribonucleoproteins. So, the congenital causes of complete atrioventricular block are either an autoimmune disease or structural heart disease. This block occurs when no atrial impulse is conducted to the ventricles [24].

Therapeutic management has changed a lot over the last decade. In the past, only a minority of patients were receiving a pacemaker; permanent cardiostimulation is indicated today as it significantly decreases mortality [24,25].

Effort capacity can be a useful tool regarding the risk of mortality, both in healthy and sick people. Effort capacity can be a useful tool regarding the risk of mortality, both in healthy and sick people. Effort capacity data in complete atrioventricular block are from many years of articles based on a few cases. Although the current literature suggests stimulating patients’ activity, sufficient data on the effect of it in these patients is not yet known [24,25,26].

### 3.6. Arrhythmias with a Genetic Substrate

#### 3.6.1. Long QT Syndrome

Long QT syndrome is an electrophysiological cardiac condition characterized by prolongation of the QT interval, a fact that indicates prolongation of the ventricular repolarization time. There are also T-wave abnormalities on the ECG that are associated with tachyarrhythmias: ventricular tachycardia or torsade de pointes. The most frequent clinical manifestation in patients with this syndrome is syncope, determined by the occurrence of the tachyarrhythmias described above. Usually, syncope occurs without warning the patient during exercises, sometimes from emotional stress. In rare cases, syncope can occur during sleep. In rare cases, torsade de pointes can degenerate into ventricular fibrillation and thus cause cardiac arrest or sudden death [17].

The congenital causes of this syndrome are Jervell-Lange-Nielsen syndrome, with autosomal recessive transmission and associated with deep neurosensory hearing loss, and RomanWard syndrome, with autosomal dominant transmission. The key distinctions between Jervell-Lange-Nielsen syndrome and Romano-Ward syndrome lie in their modes of inheritance (autosomal recessive vs. autosomal dominant) and their associated clinical features (hearing loss in Jervell-Lange-Nielsen syndrome vs. normal hearing in RomanWard syndrome). Both syndromes, however, share a common pathophysiological mechanism involving mutations in ion channels that disrupt normal cardiac electrical signaling, resulting in prolonged QT intervals and associated clinical risks. Although both syndromes share a common pathophysiological mechanism involving mutations in ion channels, autosomal dominant transmission is more common. Congenital long QT syndrome is associated with mutations in the sodium and potassium channels. Identifying the mutation involved will improve the diagnosis but will also guide future therapy [17].

Patients with LQT1 have the highest risk during physical activity due to several key factors: physiological changes (exercise increases heart rate and sympathetic nervous system activity, elevating adrenaline levels. This leads to a shorter cardiac cycle and destabilized electrical activity in the heart, raising the chance of arrhythmias); ion channel dysfunction (mutations in the KCNQ1 gene affect the potassium channel responsible for the I_Ks current, which is crucial for cardiac repolarization. Impaired function prolongs action potential duration and the QT interval, increasing arrhythmia risk under stress conditions like exercise); torsades de Pointes (TdP) (the combination of elevated heart rate and prolonged QT intervals creates conditions conducive to TdP, a dangerous form of ventricular tachycardia that can result in fainting or sudden cardiac arrest); exercise-Induced arrhythmias (the stress of exercise, coupled with increased catecholamine levels and altered ion channel function, can trigger arrhythmias in affected individuals); genetic predisposition (many patients have a family history of exercise-related sudden cardiac events, underscoring the necessity for careful management of physical activity) [20].

According to the guidelines, symptomatic patients should not participate in sports competitions. Several studies, including the latest study from Lampert et al., 2024, “Vigorous Exercise in Patients with Congenital Long QT Syndrome: Results of the Prospective, Observational, Multinational LIVE-LQTS Study”, have demonstrated that the risk of arrhythmias during sports in patients with LQTS is quite low when treated correctly. This implies that sports restrictions in this population are probably too strict [53]. Medicinal products that prolong the QT interval, electrolyte disturbances, and dehydration should be avoided. People with LQT1 are not allowed to dive in cold water because it is associated with an increased risk of arrhythmias [20].

Cardiac defibrillators are recommended for patients who have gone through cardiac arrest and those who have syncope, but it does not allow participation in sports competitions. American guidelines on participation in sports of these patients, except LQT1, are more permissive in the presence of an automatic external defibrillator. This is difficult to achieve under all circumstances [20].

#### 3.6.2. Wolff-Parkinson-White Syndrome (WPW)

WPW syndrome is a congenital preexcitation syndrome in which, in addition to the normal conduction pathway of the electrical impulse from the atria to the ventricles, there is also an abnormal pathway that conducts the impulse much faster and causes tachycardia. These can be life-threatening. From an electrocardiographic point of view, a prolonged QRS and a short PR interval are objective, as was mentioned above. Although most patients will not present arrhythmias or will be asymptomatic most of the time, there are clinical symptoms and signs specific to tachycardia, such as dizziness or palpitations. In exceptional cases, cardiac arrest can occur [11,20,32].

Sudden death in patients with preexcitation occurs during exercise. Therefore, the assessment is very important to decrease this risk. If an increased risk is identified, ablation of the accessory pathway is recommended. If the procedure is associated with increased risk, then sports activity will be discussed from patient to patient. Sports in which the potential loss of consciousness could be fatal should be discouraged. In competitive athletes or those who practice recreational physical activity who have preexcitation, it is recommended to ablate the accessory path. After ablation, low-intensity physical activity (golf, table tennis, shooting, curling, bowling, alpine skiing (recreational), basketball (adapted), and handball (adapted)) can be resumed after a week, and competitive sports (handball, basketball, football, swimming) can be resumed after one to three months [11,20,32].

In conclusion, a detailed evaluation of patients to exclude pre-excitation syndrome, ventricular arrhythmias, or structural heart disease is recommended [20]. Asymptomatic WPW with normal electrophysiological findings and no history of arrhythmia have a low risk of sudden death. The moderate to high-risk group is comprised of patients with symptomatic WPW, certain arrhythmia patterns, or those with high-risk electrophysiology (e.g., rapid conduction pathways). Critical risk patients are those with a history of arrhythmic events, those with very rapid conduction pathways, or those who experience sudden cardiac arrest during exertion.

### 3.7. Valvular Abnormalities

#### 3.7.1. Aortic Stenosis

Congenital aortic valve stenosis occurs when the aortic valve of the heart narrows and stops opening completely. Thus, blood flow from the heart to the aorta is reduced or even blocked. In common forms, the child can be asymptomatic. In the tight form, where the valvular hole is <0.5 cm^2^/m^2^, angina, sweating, and syncope may occur as symptoms and sudden death may occur in 19% of patients, especially in the presence of exercise [33].

An important criterion for establishing the therapeutic indication is the average Doppler gradient. A value of 50 mmHg or over represents the main indication for surgical intervention. There are two therapeutic methods for preserving the native valve: surgical or balloon valvulotomy. Subsequently, it will be necessary to replace the aortic valve at a later stage [19,33].

Traditionally, exercise restrictions were recommended, probably due to an increased risk of sudden death, but are now less frequent because of a more intensive approach to therapy. However, most patients are advised to avoid overexertion [19,33].

To assess hemodynamic response, patients should participate in exercise testing. High-risk individuals are identified if their blood pressure fails to increase by at least 20 mmHg during exercise or if it progressively decreases. If, after exercise, the patient develops tachycardia, then exercise should be restricted [11,20].

In patients with mild aortic stenosis, participation in low-intensity sports is generally considered safe, provided there is regular monitoring and medical oversight.

Moderate to severe aortic stenosis presents a greater risk, and patients are usually advised to avoid competitive or strenuous physical activity, as the heart’s inability to pump effectively under increased load can lead to severe complications.

Activities that cause a sudden increase in heart rate or blood pressure should be approached with caution [11,20].

#### 3.7.2. Aortic Bicuspidy

This is the most common congenital heart malformation in children and can occur either in isolation or in association with other congenital heart defects. It can also appear in association with certain genetic syndromes, such as Turner syndrome. Children can have an early onset of the disease due to valvular dysfunction or even a dilated aorta. However, most complications will appear later [35].

There are a series of phenomena that determine the appearance of symptoms and clinical signs: the degree of valvular dysfunction, the presence or absence of associated lesions, and the degree of dilation of the aorta. There are two major indications for surgical intervention in children, namely valvular stenosis and aortic regurgitation [35].

Each child is assessed individually, so they will undergo a complete assessment to determine check-up intervals, sports recommendations, and the timing of surgery [35].

It is unknown whether strenuous exercise may worsen aortic dilation or not. Today, experts approach sporting activity cautiously when the ascending aorta has dimensions above the upper limit. In the absence of aortic compromise, asymptomatic patients with preserved functional capacity can participate in low-intensity competitive and recreational physical activity: golf, table tennis, shooting, curling, bowling, alpine skiing (recreational), basketball (adapted), and handball (adapted) [20].

#### 3.7.3. Mitral Valve Prolapse

Mitral valve prolapse (MVP) is a common valvular disease characterized by excess valvular tissue in the mitral cusps. The mitral ring can be dilated, the chordae may be elongated, or there may be disordered contraction of the papillary muscles. Although often considered asymptomatic, it can progress with devastating outcomes such as mitral regurgitation, stroke, endocardial infections, and even sudden death [19,37].

Some children with prolapse feel more exhausted during physical exertion or during sports lessons compared to their peers. Clinical and laboratory investigations are used to explain the symptoms in these children. Several factors underlie the symptomatology, according to the specialized literature: reduced left ventricular filling, catecholamine and hyperadrenergic states, autonomic dysfunction, and metabolic disorders. However, the pathophysiological mechanisms remain elusive [37].

Cardiopulmonary exercise testing is a useful and essential tool in assessing exercise capacity, helping us evaluate risks and prognosis [37].

Guidelines report that MVP patients who exercise have an excellent prognosis. Asymptomatic patients with mild or moderate mitral regurgitation can participate in all competitive sports, provided there are no risk factors such as long QT syndrome, mitral annulus disjunction, or echocardiographic mechanical dispersion [20].

Mild symptomatic patients with mitral regurgitation, but without high-risk factors, can participate in low to moderate-intensity activities: golf, table tennis, shooting, curling, bowling, alpine skiing (recreational), basketball (adapted), handball (adapted), sailing, judo, karate, alpine skiing, volleyball, mid/long-distance running, dancing. Symptomatic patients should not participate in recreational or competitive activities [20].

### 3.8. Congenital Heart Defects

Individualized exercise is an effective and safe treatment method for most patients with congenital heart disease. An important predictor of prognosis and sudden cardiac death is exercise intolerance. Today, guidelines are available to determine the degree of physical activity in children with cardiac malformations based on the anatomical diagnosis. It is also necessary to obtain a complete history of the diagnosis and assess the following parameters: ventricular function, pulmonary artery pressure, aortic assessment, and arrhythmia risk. Thus, the cardiopulmonary exercise test becomes invaluable for risk assessment and predicting outcomes. Next, we describe the main congenital malformations and the recommended level of physical activity [20].

#### 3.8.1. Fallot Tetralogy

Tetralogy of Fallot is the most common cyanotic congenital heart malformation due to decreased blood oxygenation. The classic tetrad was described for the first time in 1673 by the anatomist Nicolas S. Later, in 1888, doctor Etienne F detailed the anatomical data. The classic form includes wide perimembranous ventricular septal defect, pulmonary stenosis, aortic dextroposition, and right ventricular hypertrophy [39].

Of course, there are several therapeutic strategies established by the multidisciplinary team [16,35]. Symptomatology is variable, depending on the age and severity of cyanosis, i.e., the degree of pulmonary stenosis. Cyanosis may be absent at birth, and it is important to note that tetralogy of Fallot never progresses to heart failure [39].

Since the first surgery in 1954, treatment has improved significantly so that today, it results in excellent long-term survival. However, there are common residual defects, such as right ventricular ejection tract obstruction, pulmonary regurgitation, residual ventricular septal defect, persistence of infundibular stenosis, and arrhythmias, often requiring reintervention [39].

Patients with Fallot tetralogy usually have a sedentary lifestyle that contributes to a decrease in exercise capacity. Individual counseling of patients about the practice of sports could bring significant benefits. However, children are usually restricted despite emerging evidence to support the beneficial effects of supervised and individually tailored physical activity [39].

#### 3.8.2. Atrial Septal Defect/Ventricular Septal Defect

One of the most common congenital heart conditions is an atrial septal defect, which is an abnormal connection between the two atria. This allows the mixing of systemic and pulmonary circulation. There are two criteria on which it depends on the size of the defect, namely the direction of the shunt, left-right or vice versa, and the degree. Therefore, a defect can influence morbidity and mortality depending on its significance [19,40,41].

Patients with atrial septal defects present a clinical picture that can be very variable, although they can be asymptomatic until adolescence. There are cases in which adults with wide left-right are not showing symptoms. The defect is significant if the diagnosis is made during childhood. In older children, it can cause symptoms such as tachypnea and recurrent respiratory infections [19,40,41].

In some studies, regarding the comparison between healthy and affected children, a lower maximal oxygen uptake and a lower ventilatory efficiency were observed [40].

The most commonly used therapeutic approach is the closure of the defect by means of an endovenous occluder through the Rashkind technique, an occluder shaped like an umbrella that opens in the left atrium and is attached to the wall of the hole. The optimal age for the moment of intervention is four to five years, with favorable long-term results [40].

Like atrial defect, ventricular septal defect is one of the most common congenital heart defects and represents an abnormal communication between the two ventricles [19].

Small defects close spontaneously before the age of two years at a percentage of 70–80% and are well tolerated. Those defects that allow a hemodynamically significant shunt may lead to heart failure within the first six months of life but often respond to drug therapy. Large shunts are accompanied by difficult nutrition and nutritional deficiency, and after six to twelve months, they develop pulmonary hypertension [42].

Management of ventricular septal defects varies depending on severity. Low-flow ones do not require surgical correction, but large non-restrictive ones require intervention between one to two years if the QP/QS ratio > 1.5 and pulmonary arterial resistance between 3–5 WU/m^2^. Intervention is carried out with a patch of Dacron [42].

There are inconclusive and rare data on exercise capacity in children with atrial septal or ventricular septal defects. From some studies, it was concluded that patients with closed defects have normal exercise capacity. These children are considered healthy and are encouraged to a normal level of physical activity [43]. American Heart Association (AHA, 2023) suggests that children with unrepaired defects may be restricted from intense activities such as running or competitive sports, particularly if the defect causes significant symptoms like heart murmurs, arrhythmias, or signs of heart failure.

#### 3.8.3. Transposition of the Great Arteries

It is one of the most common cyanotic heart defects, in which the two arteries, the aorta and pulmonary, are located in abnormal positions: the right ventricle gives rise to the aorta, and the left ventricle gives rise to the pulmonary artery [46].

Studies show that this population has exercise intolerance due to inadequate cardiac response (chronotropic incompetence). After surgery, several parameters have been evaluated to assess the ability of these patients to perform physical activities. Cardiopulmonary testing is performed to assess maximal exercise capacity and maximal heart rate but with continuous monitoring of respiratory gases. The workload was gradually increased, and maximum effort was reached after 10 to 15 min. Blood pressure was also recorded by sphygmomanometry. During the exercise, none of the patients showed electrocardiographic changes suggestive of myocardial ischemia. However, it is known that patients with this pathology, even after surgery, have decreased exercise capacity. Aerobic functional performance requires the integration of several systems: cardiovascular, pulmonary, and muscular, and impairment of any of these systems will lead to decreased oxygen volume and increased ventilatory equivalents [46].

According to the guidelines, they can participate in low to moderate-intensity activity—golf, table tennis, shooting, curling, bowling, alpine skiing (recreational), basketball (adapted), handball (adapted), sailing, judo, karate, alpine skiing, volleyball, mid/long distance running, dancing. It is important to remember that they must avoid long-term exercises [11,20].

#### 3.8.4. Eisenmenger Syndrome

Eisenmenger Syndrome is a complex condition that develops as a late complication of an uncorrected congenital heart defect, allowing oxygen-rich blood and oxygen-poor blood to mix. It is caused by a rare association between pulmonary arterial hypertension and congenital heart disease. Later, pulmonary hypertension will appear, and finally, Eisenmenger syndrome, with reversal of the shunt. Reversal of the shunt will produce central cyanosis and lead to secondary erythrocytosis in response to chronic hypoxemia [47,48].

There is a histological classification of pulmonary vascular changes described in 1958 by Jesse E and Donald A. of the Mayo Clinic. They considered the following grades: grade 1 is represented by medial hypertrophy of small muscular arteries and arterioles; grade 2 is represented by intimal proliferation; grade 3 is represented by progressive intimal proliferation and concentric fibrosis; grade 4 is represented by aneurysms; grade 5 is represented by glomerular proliferations; grade 6 is represented by necrosis of arterioles and arteries [47,48].

It is a multi-organ disease that includes cardiac, respiratory, neurological, haematological, gastrointestinal, endocrinological, immunological, and musculoskeletal involvement. The World Health Organization recommends advanced therapies for these patients to improve functional capacity, quality of life, and survival [47,48].

The gold standard in establishing the diagnosis, but also for determining the severity and implicitly the prognosis, is cardiac catheterization. Transthoracic ultrasound cannot diagnose pulmonary hypertension but may raise suspicion based on the values of pulmonary systolic pressures that need to be confirmed by cardiac catheterization [47,48].

Management of patients is done by a multidisciplinary team. Physical activity in these patients will lead to worsening of cyanosis due to increased right-to-left shunt. These patients are not allowed to participate in competitive sports, nor to participate in moderate and intense sports, because the risk of sudden death is increased. These children need to become aware of the limitations of physical activity, and if they perform low-intensity activity, they should monitor their heart rate so that it is <50% of the predicted maximum heart rate. Low-intensity sports such as bowling, billiards, and golf are recommended [9,11,20,47,48,49,50,51,53,54].

In conclusion, Eisenmenger syndrome has multi-organ involvement and requires follow-up by a multidisciplinary team [50].

## 4. Discussion

The aim of our study was to identify the type of physical activity allowed in children with congenital cardiac malformations and to focus on improving the long-term prognosis and quality of life for children with congenital heart disease. Given that these children are now living longer and reaching adulthood, the importance of physical activity cannot be overstated. A physically active lifestyle can prevent the development of further complications and ensure that these individuals live healthier, more fulfilling lives. Thus, encouraging physical activity while acknowledging the risks and limitations for certain patients represents the future direction for managing congenital heart defects.

The guidelines for managing physical activity in children, adolescents, and adults with congenital heart disease emphasize a balanced approach, advocating for the promotion of physical activity rather than limiting it. This represents a significant paradigm shift in pediatric cardiology. Traditionally, children with congenital heart defects were often restricted from physical exertion due to concerns about exacerbating their condition. However, recent findings and recommendations from the European Association for Cardiovascular Prevention and Rehabilitation, the European Congenital Heart and Lung Exercise Group, and the European Association of Pediatric Cardiology now encourage more dynamic exercises, even for individuals with congenital heart conditions [13].

The recognition that dynamic (aerobic) exercises provide greater cardiovascular benefits than static (anaerobic) exercises is an important advancement in patient care. Dynamic exercises stimulate the heart in a controlled way, promoting cardiac efficiency and overall physical fitness. This is particularly relevant as these patients are at higher risk of developing cardiovascular disease, and maintaining an active lifestyle is critical for preventing further complications [55,56].

In addition, the psychological and social benefits of physical activity should not be overlooked. Physical activity helps children feel more integrated into social environments and can significantly improve their quality of life. Regular exercise helps combat the stigma and isolation that children with congenital heart disease often experience. The current guidelines reinforce the message that, with proper monitoring and individualized programs, most children can participate in sports and activities at a safe level, fostering a more holistic approach to their overall health and development [55,56].

The discussion also touches on adults with congenital heart disease, who are similarly encouraged to adopt dynamic exercises. Historically, children and adults with congenital heart defects were frequently cautioned against physical exertion, but the latest evidence suggests that regular, moderate activity can significantly improve cardiovascular health in this group as well. Until recently, in the COVID era in some Eastern European countries, each individual parent gave his own recommendations to his heart disease child, taking into account personal experience but also some socio-economic aspects [57]. The literature, however, calls for more research into this area to refine the recommendations, particularly for those with complex conditions such as Eisenmenger syndrome or those who have undergone corrective surgeries.

This shift toward physical activity represents a broader recognition of the importance of overall health and well-being rather than focusing solely on the prevention of risks associated with congenital heart defects. Even for patients with potentially life-threatening conditions, movement, when done under appropriate supervision, offers tangible physical, psychological, and social benefits.

Exercise plays a significant role in improving cardiovascular health by addressing several key risk factors, including reducing blood pressure, improving cholesterol levels, and managing blood sugar levels. Studies show that regular physical activity can enhance vascular function, reduce inflammation, and increase heart function, all of which contribute to a lower risk of developing cardiovascular diseases or improve the prognosis of existing conditions. A study by Lavie et al. (2023) demonstrated that physical activity, especially aerobic and resistance exercises, significantly lowers mortality rates in individuals with cardiovascular conditions. It was found that just 150 min of moderate-intensity exercise per week had a profound impact on reducing both the incidence and progression of cardiovascular diseases [58]. While studies on children with CVD are limited, research on adults consistently supports the role of exercise in improving cardiovascular function. For example, a 2022 meta-analysis concluded that both high-intensity interval training (HIIT) and moderate-intensity aerobic exercise reduced major CVD risk factors, including hypertension and lipid profile.

Children with heart conditions require careful management of physical activity to prevent overexertion. Limiting physical activity may be necessary depending on the severity of the condition. It’s essential to tailor restrictions to the individual child, often with advice from a pediatric cardiologist. Studies generally emphasize avoiding vigorous activity during the acute phase of illness, during post-surgical recovery, or if the child experiences any arrhythmias or severe restrictions in heart function. Kourek et al. (2024) discuss how people with heart conditions benefit from a structured, age-appropriate approach to exercise that gradually increases in intensity as the child’s condition stabilizes [59].

Special attention must also be paid to children and young people with congenital or acquired diseases that affect their mobility and usual physical activity. Participants in the Paralympics are a unique category of athletes. Despite the rarity of sudden cardiac mortality in this population, it should be noted that coronary heart disease is more common in Paralympians who are unable to move. The need to increase preventive cardiology care for Paralympic athletes was brought to light by studies on the prevalence of cardiovascular system abnormalities in this group of athletes. Activities such as more frequent screening of ECGs, the use of 24-h ECG Holter monitoring, echocardiography, and cardiological care could be suggested for athletes in this group [60].

Given the potential impact on children’s health, it is crucial to develop special programs and guidelines for children with heart conditions to ensure they are safely integrated into physical activities. Tailored physical education programs, combined with parental education, can improve both the physical and psychological well-being of children with heart conditions [61]. By collaborating with pediatric cardiologists, schools can develop protocols that ensure children can safely participate in physical activities.

## 5. Conclusions

The conclusion reiterates the overwhelming benefits of an active lifestyle for children with congenital heart disease. Regular physical activity improves not only cardiovascular health but also overall physical fitness, which in turn can lower BMI, enhance muscular strength, and increase lung capacity. The improvement in cardiovascular function, which includes better heart rate control and increased endurance, leads to a better quality of life.

Additionally, the lack of extensive research and data in this area complicates the development of clear exercise programs for all types of congenital heart defects. The literature on physical activity in children with congenital heart disease remains relatively sparse, leading to inconsistencies in recommendations. While some studies have shown that these children can safely engage in exercise, there remains a degree of uncertainty, particularly regarding high-risk individuals. This highlights the need for ongoing studies that evaluate the long-term and short-term impacts of physical activity on cardiovascular health, as well as the optimal intensity, type, and frequency of exercise.

## Figures and Tables

**Table 1 children-11-01432-t001:** Overview of the congenital heart anomalies and sports indications.

Disease Name	Echo Findings	ECG Findings	Treatment	Types of Sports
Hypertrophic cardiomyopathy	Left ventricular hypertrophy, anterior systolic movement of the mitral valve, strong contraction of the left ventricle [19].	Left ventricular hypertrophy, ST segment and T wave changes, and pathological Q waves, especially in the infero-lateral leads [19].	Treatment of symptoms and prevention of sudden cardiac death. Patients with multiple risk factors should be evaluated for ICD implantation. Otherwise, amiodarone is a good alternative [20].	Patients WITHOUT risk factors can perform high-intensity sports—power sports: shot-putting, snowboarding, sprinting, water skiing, wrestling, alpine skiing [11,20] Patients WITH risk factors can perform low-intensity physical activity—golf, motor racing, horse riding, sailing, photography, ski, table tennis [11,20].
Arrhythmogenic right ventricle cardiomyopathy	Often normal; in advanced cases—dilatation of the right ventricle and formation of aneurysms; dilatation of the left ventricle may be present [16].	In precordial leads V1–V3 you can see the reversal of T waves [19]; Epsilon waves; Complete/incomplete right bundle branch block [19]; Holter ECG/24 h-extrasystoles originating in the right ventricle or unsustained ventricular tachycardia [19].	Beta Blockers (first line) for life threatening arrhythmias [19,21]; Amiodarone for symptomatic arrhythmias [19,21]; ICD for refractory/life-threatening arrhythmias [19,21].	It is not allowed to participate in sports competitions [20]; They can perform low physical activity for 150 min/week—golf, table tennis, photography [20].
Dilated cardiomyopathy	Dilation of the left and/or right ventricle [19]; Reduced global contractility function [19].	The ST segment and T wave may undergo diffuse, but non-specific changes [19]; Ventricular extrasystoles, sinus tahycardia or ventricular tahycardia may be present [19].	Treatment of heart failure [22,23]; NYHA class III and IV heart failure: cardiac resynchronization therapy and ICD [23].	People with symptoms are not encouraged to participate in competitive or recreational sports of moderate or high intensity [11,20,21]; Asymptomatic people with/without arrhythmias can participate in competitive sports: basketball, football, handball, fencing, rugby, soccer, tennis, volleyball [11,20,21].
Atrioventricular block	A wave follows the P wave during the AV block [24,25]; A multiple waves that correspond to atrial tachycardia. They appear during the complete cardiac block [24,25].	P waves are more than QRS complexes and there is no relationship between them [26]; Ventricular frequency = 30–40/min [26].	Block with narrow QRS complexes: iv atropine, and for the chronic case permanent cardiostimulation [16,20]; Block with wide QRS complexes: permanent cardiostimulation [16,20]	There are not enough studies.
Long QT syndrome	EMV is more negative in LQTS pacients [27].	The QT interval is prolonged (especially in leads DII and V5–V6) [28]; QTc > 0.50 s is associated with increased risk of torsades de pointes [28].	Hemodynamically unstable patient: electrical defibrillation [19,29]; First line: magnesium sulfate [19,29]; Not responding to MgSO_4_: temporary transvenous stimulation [19,29]; Alternatives: class IB antiarrhythmics: phenytoin and lidocaine [19,29]; Congenital long QT syndrome: beta blockers are the first choice [19,29].	Symptomatic athletes: They are not allowed to participate in competitive sports [20,29]; People with LQT1 are not allowed to dive in cold water, because it is associated with an increased risk of arrhythmias [20,29]; In the case of the ICD, the American guidelines are more permissive, provided that there are precautionary measures, such as an automatic external defibrillator [20,29].
Wolff-Parkinson-White syndrome (WPW)	M-mode echocardiography can detect the fine premature wall motion abnormalities [30].	Short PR interval; Prolonged QRS [31].	Ablation the accessory path [32].	In competitive/professional athletes with asymptomatic pre-excitation, is recommended to evaluate the risk for sudden death [20,23].
Aortic stenosis	Aortic cusps thickened, calcified, immobile [19]; Left ventricular hypertrophy [19].	Left ventricular hypertrophy [19]; The “foreign” appearance of the LV: ST-segment depression and T-wave inversion in leads DI, aVL, V5, V6—in severe aortic stenosis [19].	Preservation of the native aortic valve by either surgical valvotomy or balloon valvotomy, but often with the need for aortic valve replacement at a later stage [19,33]; Antibiotic prophylaxis of infective endocarditis [19,33].	Symptomatic stenosis: sports are not recommended [11,20]; Asymptomatic stenosis: low intensity sport: golf, table tennis, shooting, curling, bowling, alpine skiing (recreational), basketball (adapted), handball (adapted) [11,20].
Aortic Bicuspidy	Parasternal long axis: the first clue to the presence of a bicuspid valve [34]; Diastolic prolapse [34]; Systolic doming [34]; Short axis: definitive diagnosis [34]; The systolic “fishmouth” opening [34].	Left ventricular hypertrophy (the stenosis having placed a high pressure load on the left ventricle) [35].	Echocardiographic follow-up and evidence of aortic insufficiency [36]; Replacing the valve [36].	If there is no aortic stenosis/regurgitation, asymptomatic patients can participate in low-intensity activities: golf, table tennis, shooting, curling, bowling, alpine skiing (recreational), basketball (adapted), handball (adapted) [11,20,37].
Mitral valve prolapse	During the systole, the mitral sheets move to the apex < 2 mm [38]; During the diastole, maximal leaflet thickness is at least 5 mm [38].	Inversion of T waves in the inferior leads [19].	PVM with mild mitral regurgitation: the evolution is being followed; PVM with severe regurgitation: valve replacement [37].	Asymptomatic patients with mild/moderate mitral regurgitation: all sports activities, in the absence of risk factors [11,20,37]; Asymptomatic patients with severe MR, without risk factors: low-moderate intensity sports: e.g., golf, table tennis, shooting, curling, bowling, alpine skiing (recreational), basketball (adapted), handball (adapted), sailing, judo, karate, alpine skiing, volleyball, mid/long distance running, dancing [11,20,37]; Symptomatic patients with risk factors: should not participate in any kind of sports [11,20,37].
Fallot tetralogy	Ventricular septal defect [19]; Right ventricular hypertrophy [19]; Aorta straddling the septum [19]; Obstruction of the ejection path of the VD [19].	Right axial deviation [19].	Prostaglandin E to maintain PCA [19,39]; Propranolol, IV fluids, morphine, genu-pectoral position during cyanosis [19,39]; Surgical correction [19,39].	If there is an implanted ICD: activities with significant intensity must be avoided: rugby, martial arts [11,39]; If there is minor post-surgical cardiac damage: moderate activity—sailing, judo, karate, alpine skiing, volleyball, mid/long distance running, dancing [11,39]; If there is moderate heart damage: low intensity activity—golf, table tennis, shooting, curling, bowling, alpine skiing (recreational), basketball (adapted), handball (adapted) [11,39].
Atrial septal defect+ Ventricular septal defect [40,41,42]	DSA: interruption of the atrial septum that allows blood to pass between the 2 atria [19]; DSV: discontinuity of the ventricular septum that allows blood to flow [19].	DSA: right axial deviation [19]; DSV: left ventricular hypertrophy, right ventricular hypertrophy [19].	Small defect tracking [19,40]; Diuretics/ACEI: to decrease vascular resistance and volume loading in patients with large veins [19,40]; Surgical correction of wide defects and the antibiotic use [19,40,41,42].	Closed defects: normal activity [40,43].
Transpozition of the great arteries	Pulmonary arteries arising from the posterior left ventricle [44]; The aorta rising anteriorly from the right ventricle [44].	The ECG may be normal in the newborn [45]; Right ventricular hypertrophy [45]; Right axis deviation [45].	Surgically repaired through a procedure called an “arterial switch” [46].	They can participate in low to moderate intensity activity—golf, table tennis, shooting, curling, bowling, alpine skiing (recreational), basketball (adapted), handball (adapted), sailing, judo, karate, alpine skiing, volleyball, mid/long distance running, dancing [11,20]; Avoid long-term exercises [11,20].
Eisenmenger syndrome	Underlying cardiac defect [47]; Direction of intracardiac blood flow [47]; Quantification of right ventricle and pulmonary artery pressures [47]; Coexisting structural abnormalities [47]; Impaired longitudinal RV and LV strain [47].	Right axis deviation (ventricular hypertrophy) [48]; ST changes (Right ventricular or biventricular hypertrophy) [48]; Pulmonary P (Right atrial hypertrophy) [48]; A high R wave is present in V1 [48]; Deep S wave is present in V6 [48].	Antibiotics, anticoagulants, diuretics, dual endothelin receptor antagonist, iron supliements, supplemental oxygen [49,50].	Low intensity sports such as bowling, billiards, golf are recommended [11,20,50]; They have an increased risk of sudden cardiac death during intense physical activity, so moderate and intense physical activity are prohibited. Patients are also not allowed to participate in competitive sports [11,20,50].

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
