# Peer review of "Benefits of Physical Activity in Children with Cardiac Diseases—A Concise Summary for Pediatricians"

_children, 2024, doi:10.3390/children11121432_

Round 1
Reviewer 1 Report
Comments and Suggestions for Authors
Than you for your work. I have somes questions:
Line 78: Where is this guideline? "European Society of Cardiology Guidelines on Physical Activity in Patients with Congenital Malformations" ESC doens't include it. It is included info on the subject but in different guidelines.
Line 97. Why didn't you search for the FITT-VP? You only selected type, intensity and duration
Line 94: Why only selecting narrative review articles for this paper? You should screen other kind of reviews, guidelines, RCT's, etc. I know that this is a narrative review, but you should broaden your research.
Can you please identify the 55 selected articles?
Line 100: "etc." Can you please identify all the search terms. You shouldn't use the "etc".
Author Response
Dear Reviewer,
Thank you very much for evaluating our manuscript. Your recommendations and comments have helped us improve our manuscript. Here we provide the requested corrections and address the comments. The changes we have made in the manuscript are highlighted in yellow.
Line 78: Where is this guideline? "European Society of Cardiology Guidelines on Physical Activity in Patients with Congenital Malformations" ESC doens't include it. It is included info on the subject but in different guidelines.
Line 97. Why didn't you search for the FITT-VP? You only selected type, intensity and duration
Line 94: Why only selecting narrative review articles for this paper? You should screen other kind of reviews, guidelines, RCT's, etc. I know that this is a narrative review, but you should broaden your research.
Can you please identify the 55 selected articles? Line 100: "etc." Can you please identify all the search terms. You shouldn't use the "etc".
Responses:
The FITT-VP principle can certainly be adapted to children with congenital heart disease (CHD). However, our intent was to offer the pediatric cardiologist a tool to assist him/her in his collaboration with a broader team, including the sports medicine department. As pediatric cardiologists, we do not establish the workout routine of a patient; we monitor the effects. We specified this in our abstract now, to make it more clear.
For this narrative review, we searched for different studies in electronic databases, such as PubMed, ScienceDirect, and Embase. We selected 58 narrative reviews, guidelines, and original articles focusing on studies published between 2008 and 2024. We identified 58 articles that could offer us information for our review; they are written in the reference list, which has now been updated to 58 articles.
We identified all the search terms and deleted the "etc" term. Thank you for your suggestion.
We corrected the name of the guideline from line 78.
Thank you again for reviewing our manuscript.
Reviewer 2 Report
Comments and Suggestions for Authors
Dear Author;
Although physical activity is necessary and beneficial for everyone, it is even more prominent in special cases such as heart disease. This study will guide patients, clinicians, and academicians.
I would like to make some suggestions.
In the abstract, BMI is written in short and long form. It is recommended to use the long form and the abbreviation next to it where it is first used.
‘The articles were thoughtfully sorted, and duplicates were removed.’ How were duplicates eliminated? Was a system used to eliminate duplicates?
The title should be added to Table 1. In addition, it is useful to give the articles mentioned in the table one by one in the form of author-year rather than citing them by reference number.
‘3.8.2.Atrial Septal Defect . Ventricular Septal Defect’ should have a slash or and/or sign in between.
disease- or condition-specific exercise intensity selection may be more interesting and descriptive if expressed visually (by shape size and color).
References up to 54 have been numbered twice, and the journal names have not been formatted according to the citation guidelines. Please check.
Author Response
Dear Reviewer,
Thank you very much for evaluating our manuscript. Your recommendations and comments have helped us improve our manuscript. Here we provide the requested corrections and address the comments. The changes we have made in the manuscript are highlighted in yellow.
Dear Author;
Although physical activity is necessary and beneficial for everyone, it is even more prominent in special cases such as heart disease. This study will guide patients, clinicians, and academicians.
I would like to make some suggestions. In the abstract, BMI is written in short and long form. It is recommended to use the long form and the abbreviation next to it where it is first used.
‘The articles were thoughtfully sorted, and duplicates were removed.’ How were duplicates eliminated? Was a system used to eliminate duplicates?
The title should be added to Table 1. In addition, it is useful to give the articles mentioned in the table one by one in the form of author-year rather than citing them by reference number.
‘3.8.2.Atrial Septal Defect . Ventricular Septal Defect’ should have a slash or and/or sign in between.
disease- or condition-specific exercise intensity selection may be more interesting and descriptive if expressed visually (by shape size and color).
References up to 54 have been numbered twice, and the journal names have not been formatted according to the citation guidelines. Please check.
Responses: We corrected the Abstract.
No system was used to eliminate duplicates. We did so manually.
We added a title to table 1. We inserted the slash in between Atrial Septal Defect / Ventricular Septal Defect.
You are right in your advice about using shapes and colors to describe different types of sports; however, our intent was to describe which sport goes with a specific heart condition and offer a brief overview considering the specifics of each sport.
We corrected the references.
Thank you again for reviewing our manuscript,
Reviewer 3 Report
Comments and Suggestions for Authors
First of all, thank you very much for giving me the opportunity to review this important manuscript.
The number of children with congenital heart defects increases every year. Recently, these children have become more actively involved in the development of our society and occupy important places in the labor, scientific and social activities of society. However, the activity of these children in many cases ends in sudden death, which could be prevented if limits for acceptable physical activity were established.
The authors in the introduction indicated the definition and features of clinical manifestations of congenital heart pathologies.
Please add more information about the effects of exercise on improving cardiovascular disease. If data from studies of children are not available, studies of adults can be cited.
Add additional information about improving the quality of life of children with congenital heart diseases through sport exercises.
In this work you study childhood pathologies. Children by nature love to play, especially when they meet other children. Please refer to the literature for additional information on how to limit the activity of these children.
Please add additional information, is it necessary to prepare a special program for schools and kindergartens and teach parents how to deal with such children?
I would add more information on the prevalence of cardiovascular disease in Paralympic athletes. Sawczuk D, Gać P, Poręba R, Poręba M. The Prevalence of Cardiovascular Diseases in Paralympic Athletes. Healthcare (Basel). 2023 Apr 4;11(7):1027. doi: 10.3390/healthcare11071027. PMID: 37046954; PMCID: PMC10094457.
The purpose is specific and brief.
Table No. 1 is very informative, but there are some inaccuracies that need to be described in detail for clarity:
- in Hypertrophic cardiomyopathy: Patients WITHOUT risk factors and with risk factors. What risk factors are meant? What kind of low-intensity physical activity can you do while Jumping and diving?
- In dilated cardiomyopathy: “Asymptomatic people” with mild LV dysfunction/without arrhythmias can participate in competitive sport. It is better to remove “asymptomatic people” because such patients cannot be considered asymptomatic if they have mild LV symptoms.
- In WPW: In competitive/professional athletes with asymptomatic pre-excitation, is recommended to evaluate the risk for sudden death. The risk of sudden death with these diseases is always present. How to determine the risk in %? and what percentage is critical?
- In Atrial septal defect+Ventricular septal defect. Which kind od sport is allowed for children with open defect?
Write more about the adaptation of some sports, such as basketball, handball.
What are the criteria for defining low- to moderate-intensity physical activity and long-term exercise?In line 462 “Asymptomatic patients with severe mitral regurgitation”. How the disease can be asymptomatic in patients with severe mitral regurgitation?
In discussion between lines 592-597: “The recognition that dynamic (aerobic) exercises provide greater cardiovascular benefits than static (anaerobic) exercises is an important advancement in patient care. Dynamic exercises stimulate the heart in a controlled way, promoting cardiac efficiency and overall physical fitness. This is particularly relevant as these patients are at higher risk of developing cardiovascular disease, and maintaining an active lifestyle is critical for preventing further complications”. Where does this data come from? Please indicate which sources you refer to and whether this data is related to the results of your research?
In discussion between lines 598 - 603 : “In addition, the psychological and social benefits of physical activity should not be overlooked. Physical activity helps children feel more integrated into social environments and can significantly improve their quality of life. Regular exercise helps combat the stigma and isolation that children with congenital heart disease often experience. The current guidelines reinforce the message that, with proper monitoring and individualized programs, most children can participate in sports and activities at a safe level, fostering a more holistic approach to their overall health and development”. Please indicate which sources you refer to and whether this data is related to the results of your research?
Conclusion must be brief and concrete.
A additionally:
1. “The conclusion reiterates the overwhelming benefits of an active lifestyle for children with congenital heart disease. Regular physical activity improves not only cardiovascular health but also overall physical fitness, which in turn can lower BMI, enhance muscular strength, and increase lung capacity. The improvement in cardiovascular function, which includes better heart rate control and increased endurance, leads to a better quality of life. “This work examined permitted sports in children with congenital heart defects and did not demonstrate the positive impact of sports activities on the health of these children. This paragraph should be removed.
2. “An important point raised in the conclusion is the tendency for children with congenital heart conditions to become less active as they age. This decline may result from overprotective parenting or cultural and societal expectations that limit the physical capabilities of these children. While parents’ concerns are understandable, they often underestimate their children’s abilities and, in doing so, inadvertently contribute to decreased activity levels and a higher risk of cardiovascular disease in the future. It is essential for healthcare providers to communicate effectively with families, emphasizing the potential benefits of physical activity and providing specific, tailored advice”. This aspect was not considered in your study and cannot be stated in the discussion or conclusion. This paragraph would be an excellent part of the introduction.
3. “Ultimately, the focus is on improving the long-term prognosis and quality of life for children with congenital heart disease. Given that these children are now living longer and reaching adulthood, the importance of physical activity cannot be overstated. A physically active lifestyle can prevent the development of further complications and ensure that these individuals live healthier, more fulfilling lives. Thus, encouraging physical activity, while acknowledging the risks and limitations for certain patients, represents the future direction for managing congenital heart defects”. This paragraph has no connection with the results of your research. In my opinion, this paragraph will be an excellent start to your discussion.
In general, the authors have done a lot of work to identify the range of acceptable sports and the recommended intensity of exercise for patients with congenital heart pathologies. However, in reality these works are very small and lack the specificity to establish specific criteria for specialists in this field. The results covered more definitions of the operation, history, pathogenesis, and prognosis of each disease; recommendations for the use of sports exercises took up only a few sentences. In my opinion, the lack of results is also a result. I suggest that the authors emphasize the lack of serious work devoted to this problem. The effect of sports on the clinical course of congenital heart defects has not been sufficiently studied. Recommendations for sports exercises are non-specific and don’t contain systematic approaches.
Author Response
Dear Reviewer,
Thank you very much for evaluating our manuscript. Your recommendations and comments have helped us improve our manuscript. Here we provide the requested corrections and address the comments. The changes we have made in the manuscript are highlighted in yellow.
Responses:
- We added the required information in the discussion section of our article, referring to studies involving children, where possible, and studies involving adults as an alternative.
- More information on the prevalence of cardiovascular disease in Paralympic athletes has been added at your suggestion in the Discussion section.
- We corrected our mistakes from Table 1 and added the definition of adapted sports as well as classification criteria for sport intensity.
- We detailed the risk for cardiac events in WPW syndrome (section 3.6.2).
- American Heart Association (AHA, 2023) suggests that children with unrepaired atrial or ventricular septal defects may be restricted from intense activities such as running or competitive sports, particularly if the defect causes significant symptoms like heart murmurs, arrhythmias, or signs of heart failure. We added this information in the 3.8.3. section and more under Table 1.
-
Under Table 1 we also added the Criteria for Defining Low- to Moderate-Intensity Physical Activity and Long-term Exercise (AHA,2023).
- We corrected the error on line 462.
- We added references for the paragraphs that you indicated. We apologize for the error.
- We refined the conclusion section of our article at your suggestion.
Thank you again for reviewing our manuscript,
Round 2
Reviewer 1 Report
Comments and Suggestions for Authors
Line 29 page1: We selected 61 articles published 29 between 2008-2024: 55, 58 or 61 articles?
Author Response
Dear Reviewer,
After the revision of our manuscript, 61 articles were included in our review.
Thank you so much for your effort and time reviewing our work.
Reviewer 3 Report
Comments and Suggestions for Authors
Thank you very much for giving me the opportunity to review this manuscript again.
As I noted earlier, this topic is very relevant due to the lack of scientific research on sports recommendations for children with cardiovascular diseases.
The title of the article matches the content.
Many changes were made in introduction, materials, methods results that truly improved the quality and scientific soundness of the manuscript.
Errors in Table 1 have been corrected. A definition of adapted sports has been added, as well as criteria for classifying sports by intensity. Additional explanations have been added to the table description.
In my opinion, the changes in the discussion and conclusions were very successful.
More information on the prevalence of cardiovascular disease in Paralympic athletes has been added in the Discussion section.
Added additional information to the discussion section with reference to studies with children where possible and studies with adults as an alternative.
Discussion and conclusions follow logically from the results of the study and are fully consistent with the purpose of the study.
Conclusions is directly related to the data that was collected and analyzed.
Author Response
Dear Reviewer,
Thank you so much for your time and effort reviewing our work.